# Emissions from Swine Manure Treated with Current Products for Mitigation of Odors and Reduction of NH₃, H₂S, VOC, and GHG Emissions

**Baitong Chen [1], Jacek A. Koziel [1,*], Chumki Banik [1], Hantian Ma [1], Myeongseong Lee [1,2], Jisoo Wi [1,2], Zhanibek Meiirkhanuly [1], Daniel S. Andersen [1], Andrzej Białowiec [1,4] and David B. Parker [3]**

[1] Department of Agricultural and Biosystems Engineering, Iowa State University, Ames, IA 50011, USA; baitongc@iastate.edu (B.C.); cbanik@iastate.edu (C.B.); hantian@iastate.edu (H.M.); leefame@iastate.edu (M.L.); jisoowi@iastate.edu (J.W.); zhanibek@iastate.edu (Z.M.); dsa@iastate.edu (D.S.A.); andrzej.bialowiec@upwr.edu.pl (A.B.)

[2] Department of Animal Biosystems Science, Chungnam National University, Daejon 34134, Korea

[3] U.S. Department of Agriculture, Agricultural Research Service, Bushland, TX 79012, USA; david.parker@ars.usda.gov

[4] Faculty of Life Sciences and Technology, Wroclaw University of Environmental and Life Sciences, 37a Chełmonskiego Str., 51-630 Wroclaw, Poland

[*] Correspondence: koziel@iastate.edu; Tel.: +1-515-294-4206

**Abstract:** Odor and gaseous emissions from the swine industry are of concern for the wellbeing of humans and livestock. Additives applied to the swine manure surface are popular, marketed products to solve this problem and relatively inexpensive and easy for farmers to use. There is no scientific data evaluating the effectiveness of many of these products. We evaluated 12 manure additive products that are currently being marketed on their effectiveness in mitigating odor and gaseous emissions from swine manure. We used a pilot-scale system simulating the storage of swine manure with a controlled ventilation of headspace and periodic addition of manure. This dataset contains measured concentrations and estimated emissions of target gases in manure headspace above treated and untreated swine manure. These include ammonia (NH₃), hydrogen sulfide (H₂S), greenhouse gases (CO₂, CH₄, and N₂O), volatile organic compounds (VOC), and odor. The experiment to test each manure additive product lasted for two months; the measurements of NH₃ and H₂S were completed twice a week; others were conducted weekly. The manure for each test was collected from three different farms in central Iowa to provide the necessary variety in stored swine manure properties. This dataset is useful for further analyses of gaseous emissions from swine manure under simulated storage conditions and for performance comparison of marketed products for the mitigation of gaseous emissions. Ultimately, swine farmers, the regulatory community, and the public need to have scientific data informing decisions about the usefulness of manure additives.

**Keywords:** odor mitigation; sustainable agriculture; air quality; gaseous emissions; environmental technologies; animal production systems; swine manure; waste management

---

## 1. Summary

The United States is one of the top three pork-producing countries, and Iowa is the biggest pork-producing state. In Iowa, the most common structure for manure storage is a deep pit under a barn, as shown in Figure 1. The deep pit under the slatted floor often stores manure for a year, which is then applied as a fertilizer on the fields (in the fall after harvest). This commodity generates

profits and jobs for local communities, but also generates odorous chemical gases that affect the air quality in the surrounding communities. These unwanted gaseous emissions are one of the biggest concerns in the pork industry. Ammonia ($NH_3$) and hydrogen sulfide ($H_2S$) are extremely harmful to the health of humans and animals, especially during the manure pump-out season. During this time, there is always news about people or livestock fainting, or fatalities due to exposure from $NH_3$ and $H_2S$ [1]. Volatile organic compounds (VOCs) are the main generators of odor. Small particle matters in the air will carry these odorous VOCs miles away and affect the living environments of farming communities [2]. Greenhouse gases (GHG) will accumulate in the atmosphere and cause changes in the climate. The manure additive is one of many ways to potentially mitigate these emissions. It is considered to be relatively inexpensive compared with other methods and easy to apply without changing the swine farm's current structures [3].

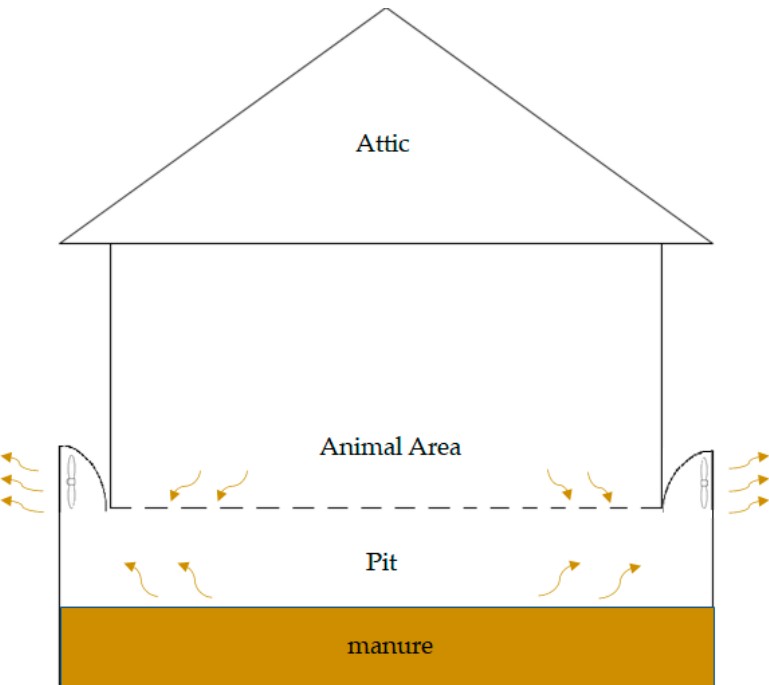

**Figure 1.** The schematic of a deep pit swine farm structure used in Iowa.

For these trials, the manure was collected from three different farms in central Iowa. Twelve on-market products including Triune, Confine N, Manure Master Plus, Sufi-dox$_{dry}$, Waste Away, Enviro Lagoon, More Than Manure, Oxydol, Sludge Away, Penergetic G, Manure Magic, and LLMO-SST were tested for a two-month period. The information on the company name, purpose, mode of action, and recommended dosages for all 12 products are given in Table A1.

$NH_3$ and $H_2S$ were measured with a Drager X-am 5600 portable gas analyzer and OMS-300. VOCs were collected using a solid-phase microextraction and characterized by a gas chromatography-mass spectrometry (GC-MS). Greenhouse gases were analyzed by using a GHG-GC equipped with a flame ionization detector (FID) and electron capture detector (ECD). Odor concentration was evaluated with the AC'SCENT International Olfactometer.

The National Pork Board funded this project to evaluate the effectiveness of on-market products for the mitigation of odors and reduction of $NH_3$, $H_2S$, VOCs, and GHG. This dataset can provide the science-based, comprehensive measurements of gaseous emissions from swine manure. It can also be used to gain a better understanding of the relationship between gaseous emissions and manure properties. This dataset could potentially provide a correlation among the gaseous emissions (i.e., how changes in one gas emission change other gases emissions). The last comprehensive research

on 35 marketed manure additives in a pilot-scale with manure storage simulators was published in 2001 [4].

## 2. Data Description

This dataset is in a spreadsheet format, which includes $NH_3$, $H_2S$, VOCs, and GHG emissions from untreated swine manure (control) and swine manure treated with manure additives for two months. There is a total of six spreadsheets. The first spreadsheet gives the abbreviations of the words used in the document, dimensions of the manure simulators, and flowrates. The second spreadsheet provides information regarding the properties of the manure for the four trials conducted in this research. The third spreadsheet shows the flux and concentrations of $NH_3$ and $H_2S$. The fourth spreadsheet displays the concentrations and flux of greenhouse gases: $CO_2$, $CH_4$, and $N_2O$. The fifth spreadsheet relates to odor concentration, and the last spreadsheet is the peak area counts of VOCs.

### 2.1. Manure Properties

The spreadsheet on manure properties contains the percent of total solids (%TS) (column C & M), the percent of volatile solids (%VS) (column D & N), total Kjeldahl nitrogen in mg per liter (TKN) (column E & O), the nitrogen content of the ammonium ion in mg per liter ($NH_4$-N) (column F & P), and total phosphorous in mg per liter (TP) (column G & Q). The manure properties given are from before the application of any treatment and after eight weeks for each of the four trials. Before the treatment, manure properties are examined separately due to different manures coming from different farms. After eight weeks of trial, the manure properties for both control and treated manure are obtained. An example of the manure properties are summarized in 'Manure Additives.xlsx' spreadsheet (Supplementary Materials, Sheet 'Manure Properties') as shown in Table 1.

**Table 1.** An example of the manure properties summarized in 'Manure Additives.xlsx' spreadsheet (Supplementary Materials, Sheet 'Manure Properties').

| | Trial 1 Manure after Eight Weeks | | TS | VS | TKN | $NH_4$-N | TP |
|---|---|---|---|---|---|---|---|
| Simulator # | Manure Sources | Treatment | % | % | mg·L$^{-1}$ | mg·L$^{-1}$ | mg·L$^{-1}$ |
| 1 | Deep pit 1 | Triune | 5.9 | 4.7 | 5500 | 4700 | 2376 |
| 2 | Outdoor | Confine | 3.6 | 3 | 3500 | 3200 | 1944 |
| 3 | Deep pit 2 | Confine | 7 | 5.7 | 4600 | 4000 | 3240 |
| 4 | Deep pit 2 | Triune | 6.9 | 5.6 | 4500 | 3800 | 3348 |
| 5 | Outdoor | MMP | 3.7 | 2.9 | 3100 | 2800 | 1620 |
| 6 | Deep pit 2 | MMP | 7 | 5.8 | 4600 | 3900 | 3024 |
| 7 | Deep pit 1 | Sulfi | 5.9 | 4.8 | 5400 | 4600 | 2592 |
| 8 | Deep pit 2 | Sulfi | 6.8 | 5.5 | 4800 | 4100 | 3456 |
| 9 | Deep pit 2 | Control | 6.9 | 5.7 | 4500 | 4000 | 2808 |
| 10 | Outdoor | Sulfi | 3.7 | 3 | 6100 | 5500 | 2160 |
| 11 | Deep pit 1 | Control | 5.7 | 4.6 | 5300 | 4500 | 2808 |
| 12 | Deep pit 1 | MMP | 5.6 | 4.6 | 5000 | 4300 | 2160 |
| 13 | Outdoor | Control | 3.8 | 3 | 3000 | 2700 | 1728 |
| 14 | Outdoor | Triune | 3.8 | 3 | 3400 | 3100 | 2160 |
| 15 | Deep pit 1 | Confine | 5.8 | 4.6 | 5200 | 4500 | 2592 |

### 2.2. $NH_3$ and $H_2S$

In the spreadsheet pertaining to $NH_3$ and $H_2S$, both concentrations are given in ppm (column F & G). The flux is given in mg per hour per square meter (mg·h$^{-1}$·m$^{-2}$) (column H & I). Concentrations were directly measured using Drager X-am 5600 and OMS-300; the flux was calculated from the concentrations by using the room temperature and pressure in central Iowa, and the conditions of the manure simulators such as flowrate and headspace. Each measurement of $NH_3$ and $H_2S$ has a corresponding manure simulator number (column A), trial number (column B), treatment (column C),

the farm of manure origin (column D), and the number of days before or after the application of treatments (column E). If the day is a negative number, measurements were also completed before any treatment application. There was no $H_2S$ concentration detected in the last trial. An example of concentrations and fluxes of $NH_3$ and $H_2S$ are summarized in 'Manure Additives.xlsx' spreadsheet (Supplementary Materials, Sheet '$NH_3$ & $H_2S$') as shown in Table 2.

**Table 2.** An example of concentrations and fluxes of $NH_3$ and $H_2S$ summarized in 'Manure Additives.xlsx' spreadsheet (Supplementary Materials, Sheet '$NH_3$ & $H_2S$').

| Simulator # | Trial | Treatment | Manure Sources | Time (day)* | $H_2S$ (ppm) | $NH_3$ (ppm) | $H_2S$ Flux $(mg \cdot h^{-1} \cdot m^{-2})$ | $NH_3$ Flux $(mg \cdot h^{-1} \cdot m^{-2})$ |
|---|---|---|---|---|---|---|---|---|
| 1 | 1 | Triune | Deep pit 1 | −6 | 0.00 | 55.28 | 0.00 | 111.76 |
| 2 | 1 | Confine | Outdoor storage | −6 | 0.50 | 41.48 | 0.91 | 75.80 |
| 3 | 1 | Confine | Deep pit 2 | −6 | 0.50 | 72.85 | 0.84 | 122.84 |
| 4 | 1 | Triune | Deep pit 2 | −6 | 0.00 | 69.51 | 0.00 | 136.74 |
| 5 | 1 | MMP | Outdoor storage | −6 | 0.60 | 54.84 | 1.08 | 98.50 |
| 6 | 1 | MMP | Deep pit 2 | −6 | 0.60 | 71.24 | 1.09 | 129.11 |
| 7 | 1 | Sulfdox | Deep pit 1 | −6 | 0.40 | 130.48 | 0.73 | 238.73 |
| 8 | 1 | Sulfdox | Deep pit 2 | −6 | 0.00 | 66.35 | 0.00 | 128.20 |
| 9 | 1 | Control | Deep pit 2 | −6 | 0.70 | 105.99 | 1.32 | 200.38 |
| 10 | 1 | Sulfdox | Outdoor storage | −6 | 0.60 | 17.47 | 1.11 | 32.40 |
| 11 | 1 | Control | Deep pit 1 | −6 | 0.50 | 96.39 | 0.88 | 170.47 |
| 12 | 1 | MMP | Deep pit 1 | −6 | 0.00 | 118.53 | 0.00 | 225.80 |
| 13 | 1 | Control | Outdoor storage | −6 | 0.60 | 44.85 | 1.13 | 84.68 |
| 14 | 1 | Triune | Outdoor storage | −6 | 0.00 | 50.54 | 0.00 | 98.63 |
| 15 | 1 | Confine | Deep pit 1 | −6 | 0.50 | 97.70 | 0.93 | 181.10 |

* Days prior to the application of manure additive are marked with negative (−) sign.

## 2.3. GHG

Greenhouse gases concentrations are given in ppm (column F, G, H), with the flux in $mg/h/m^2$ (column I, J, K) for $CO_2$, $CH_4$, and $N_2O$. Additionally given are the corresponding manure simulators (column A), trial numbers (column B), treatments (column C), manure sources (column D), and the number of days before or after applying treatments (column E). The GHG-GC analyzed each gas by its output peak area. By using standard calibration curves, the concentrations were calculated into parts per million (PPM). The same method was used to calculate the flux from the concentrations. An example of concentrations and fluxes of greenhouse gases are summarized in 'Manure Additives.xlsx' spreadsheet (Supplementary Materials, Sheet 'GHG'), as shown in Table 3.

**Table 3.** An example of concentrations and fluxes of greenhouse gases (GHG) summarized in 'Manure Additives.xlsx' spreadsheet (Supplementary Materials, Sheet 'GHG').

| Simulator # | Trial | Treatment | Manure Sources | Time (day) | $CO_2$ (ppm) | $CH_4$ (ppm) | $N_2O$ (ppm) | $CO_2$ Flux ($mg \cdot h^{-1} \cdot m^{-2}$) | $CH_4$ ($mg \cdot h^{-1} \cdot m^{-2}$) | $N_2O$ ($mg \cdot h^{-1} \cdot m^{-2}$) |
|---|---|---|---|---|---|---|---|---|---|---|
| 1 | 1 | Triune | Deep pit 1 | 2 | 799 | 42 | 0.226 | 3777 | 74 | 1.068 |
| 2 | 1 | Confine | Outdoor storage | 2 | 666 | 34 | 0.220 | 3148 | 60 | 1.040 |
| 3 | 1 | Confine | Deep pit 2 | 2 | 578 | 10 | 0.220 | 2732 | 18 | 1.040 |
| 4 | 1 | Triune | Deep pit 2 | 2 | 603 | 40 | 0.219 | 2851 | 70 | 1.035 |
| 5 | 1 | MMP | Outdoor storage | 2 | 518 | 30 | 0.205 | 2449 | 53 | 0.969 |
| 6 | 1 | MMP | Deep pit 2 | 2 | 438 | 16 | 0.216 | 2071 | 28 | 1.021 |
| 7 | 1 | Sulfdox | Deep pit 1 | 2 | 1223 | 94 | 0.207 | 5781 | 166 | 0.979 |
| 8 | 1 | Sulfdox | Deep pit 2 | 2 | 647 | 14 | 0.195 | 3058 | 25 | 0.922 |
| 9 | 1 | Control | Deep pit 2 | 2 | 561 | 15 | 0.242 | 2652 | 26 | 1.144 |
| 10 | 1 | Sulfdox | Outdoor storage | 2 | 450 | 47 | 0.190 | 2127 | 83 | 0.898 |
| 11 | 1 | Control | Deep pit 1 | 2 | 918 | 48 | 0.244 | 4340 | 85 | 1.154 |
| 12 | 1 | MMP | Deep pit 1 | 2 | 1047 | 68 | 0.207 | 4949 | 120 | 0.979 |
| 13 | 1 | Control | Outdoor storage | 2 | 433 | 30 | 0.225 | 2047 | 53 | 1.064 |
| 14 | 1 | Triune | Outdoor storage | 2 | 582 | 39 | 0.207 | 2751 | 69 | 0.979 |
| 15 | 1 | Confine | Deep pit 1 | 2 | 706 | 52 | 0.221 | 3337 | 92 | 1.045 |

*2.4. Odor*

The spreadsheet for odor shows the odor concentration (column F), manure simulator number (column A), trial number (column B), treatments (column C), farm (column D), and the number of days before or after treatments (column E). The odor concentration was analyzed based on the European Odor Unit Standard methods and AC'SCENT International Olfactometer. An example of the odor concentration is summarized in 'Manure Additives.xlsx' spreadsheet (Supplementary Materials, Sheet 'Odor') as shown in Table 4.

**Table 4.** An example of the odor concentration summarized in 'Manure Additives.xlsx' spreadsheet (Supplementary Materials, Sheet 'Odor').

| Simulator # | Trial | Treatment | Manure Source | Time (day) | Odor Concentration (OU·m$^{-3}$) |
|---|---|---|---|---|---|
| 1 | 1 | Triune | Deep pit 1 | 3 | 420 |
| 2 | 1 | Confine | Outdoor storage | 3 | 263 |
| 3 | 1 | Confine | Deep pit 2 | 3 | 960 |
| 4 | 1 | Triune | Deep pit 2 | 3 | 2262 |
| 5 | 1 | MMP | Outdoor storage | 3 | 479 |
| 6 | 1 | MMP | Deep pit 2 | 3 | 520 |
| 7 | 1 | Sulfdox | Deep pit 1 | 3 | 449 |
| 8 | 1 | Sulfdox | Deep pit 2 | 3 | 278 |
| 9 | 1 | Control | Deep pit 2 | 3 | 691 |
| 10 | 1 | Sulfdox | Outdoor storage | 3 | 2900 |
| 11 | 1 | Control | Deep pit 1 | 3 | 524 |
| 12 | 1 | MMP | Deep pit 1 | 3 | 1278 |
| 13 | 1 | Control | Outdoor storage | 3 | 3412 |
| 14 | 1 | Triune | Outdoor storage | 3 | 342 |
| 15 | 1 | Confine | Deep pit 1 | 3 | 797 |

*2.5. VOCs*

The last spreadsheet contains the results for each targeted VOC in all four trials. The spreadsheet summarizes the data collected on the volatile organic compounds (VOCs) emitted from manures collected from three different sources (deep pit 1, deep pit 2, and outdoor storage or deep pit 3), all with different treatments and one control for each type of manure. The rows are providing the information on the manure simulator unit (column A), trial number (column B), type of treatment applied (column C), farm type for the manure collected (column D), the time before or after the treatments applied in weeks, if the week is appeared to be negative, then it means weeks before applying any manure additive (column E), volatile organic compound name (column F) for the particular VOC for the selected ion monitoring (SIM) method, ion selected (column G), chemical abstract service (CAS) number (column H), retention time (column I), and relative abundance or peak area count (column J) of that particular VOC. A total of five phenolic, three sulfidic, and three volatile fatty acids have been recorded as potent odorous compounds in the manure headspace for the first three trials. The last trial only detected five phenolics. An example of the relative abundance of VOCs is summarized in 'Manure Additives.xlsx' spreadsheet (Supplementary Materials, Sheet 'VOCs') as shown in Table 5.

**Table 5.** An example of the relative abundance of volatile organic compounds (VOCs) summarized in 'Manure Additives.xlsx' spreadsheet (Supplementary Materials, Sheet 'VOCs').

| Simulator # | Trial | Treatment | Manure Source | Time (week)* | Compound Name | Ion Selected for Abundance | CAS # | GC Column Retention Time (min) | Peak Area Count (arbitrary units) |
|---|---|---|---|---|---|---|---|---|---|
| 1 | 1 | Triune | Deep pit 1 | −1 | Skatole | 130 | 83-34-1 | 29.16 | 1,289,970 |
| 2 | 1 | Confine | Outdoor storage | −1 | Skatole | 130 | 83-34-1 | 29.16 | 1,208,680 |
| 3 | 1 | Confine | Deep pit 2 | −1 | Skatole | 130 | 83-34-1 | 29.16 | 553,325 |
| 4 | 1 | Triune | Deep pit 2 | −1 | Skatole | 130 | 83-34-1 | 29.16 | 397,306 |
| 5 | 1 | MMP | Outdoor storage | −1 | Skatole | 130 | 83-34-1 | 29.16 | 1,482,050 |
| 6 | 1 | MMP | Deep pit 2 | −1 | Skatole | 130 | 83-34-1 | 29.16 | 578,592 |
| 7 | 1 | Sulfdox | Deep pit 1 | −1 | Skatole | 130 | 83-34-1 | 29.16 | 2,802,650 |
| 8 | 1 | Sulfdox | Deep pit 2 | −1 | Skatole | 130 | 83-34-1 | 29.16 | 747,174 |
| 9 | 1 | Control | Deep pit 2 | −1 | Skatole | 130 | 83-34-1 | 29.16 | 1,172,744 |
| 10 | 1 | Sulfdox | Outdoor storage | −1 | Skatole | 130 | 83-34-1 | 29.16 | 5,205,639 |
| 11 | 1 | Control | Deep pit 1 | −1 | Skatole | 130 | 83-34-1 | 29.16 | 704,348 |
| 12 | 1 | MMP | Deep pit 1 | −1 | Skatole | 130 | 83-34-1 | 29.16 | 1,529,776 |
| 13 | 1 | Control | Outdoor storage | −1 | Skatole | 130 | 83-34-1 | 29.16 | 364,090 |
| 14 | 1 | Triune | Outdoor storage | −1 | Skatole | 130 | 83-34-1 | 29.16 | 1,620,530 |
| 15 | 1 | Confine | Deep pit 1 | −1 | Skatole | 130 | 83-34-1 | 29.16 | 1,887,679 |

* Weeks prior to the application of manure additive are marked with negative (−) sign.

### 3. Methods

The pilot-scale setup (Figure 2, Figure A1) simulates manure storage and ventilation of deep pits under swine barns (Figure 1) [5]. There are 15 manure simulator units; each unit has a height of 1.22 m (4 feet) and a diameter of 0.38 m (15 inches). Initially, 74.6 L of manure from three different farms in central Iowa was pumped into each storage unit. Biweekly, 9.5 L of manure was added into each simulator unit to simulate storage at a real farm. Flowrates were controlled by FL-3839ST rotameters (Omega Engineering Inc, USA) with ± 2% full-scale accuracy and ± 0.25% full-scale repeatability to achieve 7.5 headspace air exchange per hour (ACH) and adjusted to 7.5 ACH when fresh manure was added (Figure 2, Figure A1). Since the source of the inlet air was indoors, the temperature of manure was close to the room temperature. The room has temperature control (and an air conditioning and a space heater) to keep the room around 14 to 21 °C. In north central Iowa, measured temperatures of the manure pit can be as cold as 9 °C in the winter and as warm as 22 °C in the summer [6]. Trial 1 was conducted from December to February, and the mean room temperature for trial 1 was 15 °C. Trial 2 was conducted from April to May, and the mean room temperature was 16.2 °C. Trial 3 was conducted from July to September, and the mean temperature was 21.4 °C. The last trial was conducted from October to December, and the mean room temperature was 16.6 °C. The application rate and methods of manure additives were followed by each product description (Table A1). Some additive products contain instruction for an application rate based on the manure pit surface area. We converted the per surface area instructions into manure volume by using the standard (in US) space area per head and the average production of manure per head.

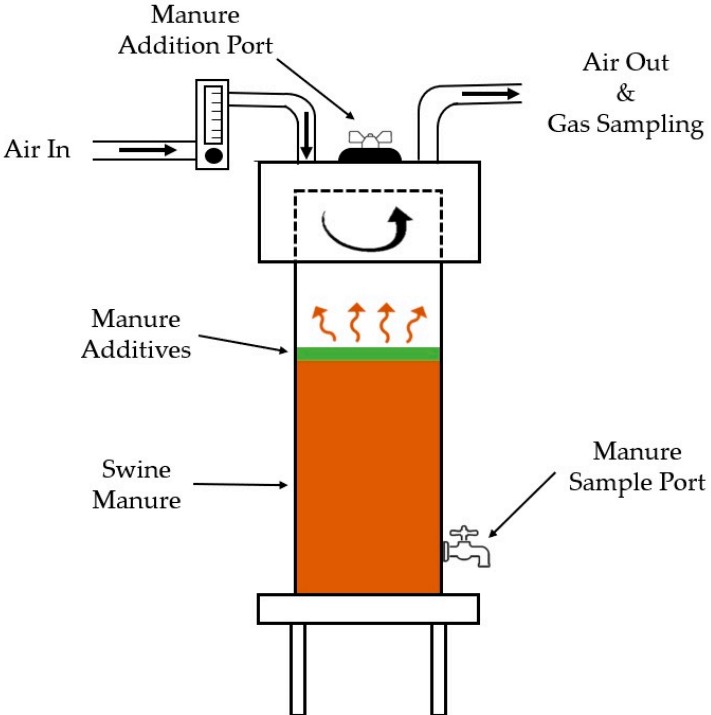

**Figure 2.** Schematic of manure storage simulating the deep pit swine barn. A total of 15 storage simulators were used, facilitating tests of four manure additives in n = 3 replicates over eight-week trials.

Swine manure properties change with time, even for manures from the same farm. This is due to many farm management and environmental factors. It is also agreed among the researchers working with the manure that a 'perfect' replicate of manure is difficult (if not impossible) to make (likely due to the unpredictable activity of microbes and lack of control). This leads to statistically treating manure as a 'biological' material that is inherently variable. The manufacturer's directions tested in this research do not consider 'manure' as being site-specific, i.e., they can be applied to all types and

sources of swine manure. There were four trials of experiments conducted. The first two trials tested four manure additive products with n = 3 replicates. The third trial tested three products with n = 3 replicates. The last trial tested only one product with n = 3 replicates. All trials lasted for eight weeks.

### 3.1. Odor

The air samples were collected by using a Vac-U-Chamber (SKC Inc., Eighty-Four, PA, USA) and 10 L of Tedlar sample bags. Every time before collecting the samples, Tedlar sample bags were flushed with air and vacuumed out. Odor samples were analyzed by the AC'SCENT International Olfactometer (St. Croix Sensory Inc., Stillwater, MN, USA) using dynamic triangular forced-choice methods, and all samples were analyzed within 24 h. Each sample was then analyzed twice by four panelists [7]. The Olfactometer is shown in Figure A2.

### 3.2. $NH_3$ and $H_2S$

$NH_3$ and $H_2S$ concentrations were measured using a Dräeger X-am 5600 portable gas analyzer (Luebeck, Germany) (Figure A3) and OMS-300 (Smart Control and Sensing Inc., Daejeon, Rep. of Korea) (Figure A4). Dräegers equipped with a $NH_3$ XXS sensor with 0–300 ppm and $H_2S$ XXS sensor with 0–100 ppm were calibrated using standard gases (298 ppm $NH_3$ and 50.4 ppm $H_2S$) [8], and calibration curves were established to correct the results. OMS-300, which can be used in a real-time monitoring system for gases, was equipped with electrochemical gas sensors $NH_3$/CR-1000 and $H_2S$/C-50 (Wallisenllen, Switzerland) [9]. Both gas sensors were calibrated using standard gases before the experiment.

### 3.3. GHG

Greenhouse gases were collected in 5.9 mL Exetainer vials (Labco Limited, London, UK) using syringes. Samples were analyzed with a GC equipped with FID and ECD (SRI Instruments, Torrance, CA, USA) as shown in Figure A5. Exetainer vials were precleaned by flushing with Helium gas (UHP 300) and vacuumed with a pump. Before each GHG analysis, standard calibration curves were made by using 1005 and 4010 ppm of $CO_2$, 10 and 20 ppm of $CH_4$, and 0.1 and 1 ppm of $N_2O$ (Air Liquide America, Plumsteadville, PA, USA) [5].

### 3.4. VOCs

The air samples were collected in 1 L gas sampling glass bulbs (Supelco) using portable sampling pumps. Each gas sampling glass bulb was flushed with DI water and baked in the oven overnight before the gas sampling day. Immediately after each sample collection, 1 µL of internal standard (100 ppm Hexane) was injected into each gas sampling glass bulb onsite to check for leakage after being analyzed by the GC-MS, as shown in Figure A6. The gas bulbs were stored in a cooler, brought to the laboratory, and were analyzed using a multidimensional GC-MS within 24 h of collection.

The gas samples were analyzed using the Agilent 6890 (Agilent Technologies, Santa Clara, CA, USA) customized multidimensional gas chromatograph (Microanalytics, a part of volatile analysis Corporation, Round Rock, TX, USA). On the chromatography part, there were two capillary columns. A 5% phenyl polysilphenylene-siloxane (30 m × 0.53 mm inner diameter × 0.5 µm thickness, Trajan Scientific, Austin, TX, USA) column, followed by a polar column bonded polyethylene glycol in a Sol-Gel matrix (30 m × 0.53 mm inner diameter × 0.5 µm thickness, Trajan Scientific, Austin, TX, USA). The first column was fixed with a restrictor precolumn (connected to an injector), and the midpoint of the two columns was maintained at a constant pressure of 0.39 atm by a pneumatic switch. For this research, the multidimensional capability of the GC was not needed for the separations of target VOCs. Yet, the unusual coupling of columns was still in place out of convenience, and all effluent from the nonpolar column was directed to the polar column. Ultra-high pure He (99.999%, Airgas, Des Moines, IA, USA) was used as the carrier gas. The discharge from the second column was simultaneously directed to a single quadrupole Agilent mass spectrometer (MS) (Model 5973N, Santa Clara, CA, USA).

The GC oven temperature was programmed at an initial 40 °C for 3 min, followed by an increase to 240 °C at a rate of 7 °C/min, where it was maintained for 8.43 min. The quadrupole MS was set to an electron ionization mode, with ionization energy of 70 eV during operation and a full scan range of 34 to 350 m/z. The system automation software was Multitrax v. 6.00.1 (Microanalystics, Round Rock, TX, USA), and the data acquisition software was ChemStation E.01.01.335 (Agilent Technologies, Santa Clara, CA, USA). The multidimensional GC-MS schematic and a comprehensive description were published elsewhere [10].

Gas samples from the gas bulbs were extracted using a 2 cm 50/30 µm DVB/PDMS/Carboxen fused silica SPME fiber (57248-U, Supelco, Bellefonte, PA, USA) at lab temperature (23–24 °C) for 50 min. The SPME fibers which were loaded with the VOCs from the samples were injected to the GC injector and heated at 260 °C to thermally desorb the VOCs to the GC columns. Then, the fibers were separated by the GC column and analyzed by the mass spectrometer to measure the relative abundance of each targeted VOC, which followed the similar protocols from previous studies [11,12]. The samples were analyzed in the SIM mode because of its higher sensitivity and lower detection limit as opposed to the total scan mode. To identify the compounds, the NIST mass spectral library was (with at least 80% spectral match) used in this study. Further, to verify the retention time of the VOCs studied, pure standards of all 15 VOCs were analyzed and calibrated [13,14]. The VOC concentrations were not quantified. A surrogate metric of VOC abundance (measured with peak area counts) was used to assess the performance of manure additives to mitigate emissions from swine manure. The dataset could be used to estimate the mitigation effect by comparing the VOC abundance in the treatment and control.

### 3.5. Manure Properties

The total solids (TS) and volatile solids (VS) concentrations were measured using standard methods 2540 B and 2540 E [15]. The pH values were measured using an Accumet Basic AB15 Plus pH meter and Accumet 13-620-285 pH probe. Ammonia concentrations were measured using standard methods 4500-NH3-B Preliminary Distillation Step and 4500-NH3-C Titrimetric Method, with 0.1 N HCl as the titrant instead of sulfuric acid. The dissolved reactive phosphorus concentration was analyzed using standard method 4500-P E [15]. Total Kjeldahl nitrogen (TKN) concentrations were measured using the distillation method and titration method described for ammonia, but with digestion using standard method 2001.11 from AOAC Official Methods. Total Phosphorus (TP) was measured using the AOAC method 965.17, starting with digestion with hydrochloric and nitric acid.

## 4. User Notes

Supplementary Materials contain an Excel spreadsheet (Manure additives.xlsx). The spreadsheet includes a total of six sheets.

| Excel Sheets | Content |
| --- | --- |
| Information | The code name of manure source, abbreviations of manure product, and parameters of the manure simulator. |
| Manure Properties | Manure properties have four sets of tables for four trials of experiments with percent total solids (TS), percent volatile solids (VS), total nitrogen content (TKN), the nitrogen content of ammonium (NH4-N), and total phosphorous (TP). |
| $NH_3$ & $H_2S$ | This spreadsheet has the measured concentrations and fluxes of ammonia and hydrogen sulfide for each manure source and manure additive. |
| GHG | This spreadsheet summarizes measured concentrations and fluxes for carbon dioxide ($CO_2$), methane ($CH_4$), and nitrous oxide ($N_2O$). |
| Odor | This spreadsheet summarizes the odor concentration measurement using a dilution olfactometer. |
| VOCs | The spreadsheet contains the information of volatile organic compounds (VOCs), i.e., compound names, selected ion used to identify the compound, CAS number, GC column retention time, and peak area counts. |

**Supplementary Materials:** The following are available online at http://www.mdpi.com/2306-5729/5/2/54/s1. All data is presented in the Excel spreadsheet (Manure Additives.xlsx).

**Author Contributions:** Conceptualization, J.A.K. and D.S.A.; methodology, B.C., J.A.K., and D.S.A.; validation, J.A.K. and D.S.A.; formal analysis, B.C.; investigation, B.C., H.M., M.L., Z.M., J.W., and C.B.; resources, J.A.K. and D.S.A.; data curation, B.C. and J.K.; writing—original draft preparation, B.C.; writing—review and editing, J.K. and A.B.; visualization, B.C. and M.L.; supervision, J.K.; project administration, J.K., D.B.P, and D.A.; funding acquisition, J.K., D.B.P., and D.A. All authors have read and agreed to the published version of the manuscript.

**Funding:** This research was partially funded by the National Pork Board and the Indiana Pork Producers Association, grant NBP-17-158: Evaluation of current products for use in deep pit swine manure storage structures for mitigation of odors and reduction of $NH_3$, $H_2S$, and VOC emissions from stored swine manure (2018-2020, PI J.K.). In addition, this research was partially supported by the Iowa Agriculture and Home Economics Experiment Station, Ames, Iowa. Project no. IOW05556 (Future Challenges in Animal Production Systems: Seeking Solutions through Focused Facilitation) sponsored by Hatch Act and State of Iowa funds. The authors would like to thank the Ministry of Education and Science of the Republic of Kazakhstan for supporting Zhanibek Meiirkhanuly with an M.S. study scholarship via the Bolashak Program. Authors would also like to thank the Fulbright Foundation for funding the project titled "Research on pollutants emission from Carbonized Refuse Derived Fuel into the environment," completed by A.B. at the Iowa State University.

**Acknowledgments:** The authors are grateful to the Iowa State University Honors Program (Ana DiSpirito, Freshmen Honors) and George Washington Carver Summer Internship Program (Lizbeth Plaza-Torres), Danielle Wrzesinski, Blake Fonken, and Wyatt Murphy for their help with manure hauling and odor analyses. The preliminary findings from the testing of the first set of eight manure additives were documented in the M.S. thesis by the first author (Baitong Chen, 2019) [16].

**Conflicts of Interest:** The authors declare no conflict of interest. The funders had no role in the design of the study; in the collection, analyses, or interpretation of data; in the writing of the manuscript, or in the decision to publish the results.

## Appendix A

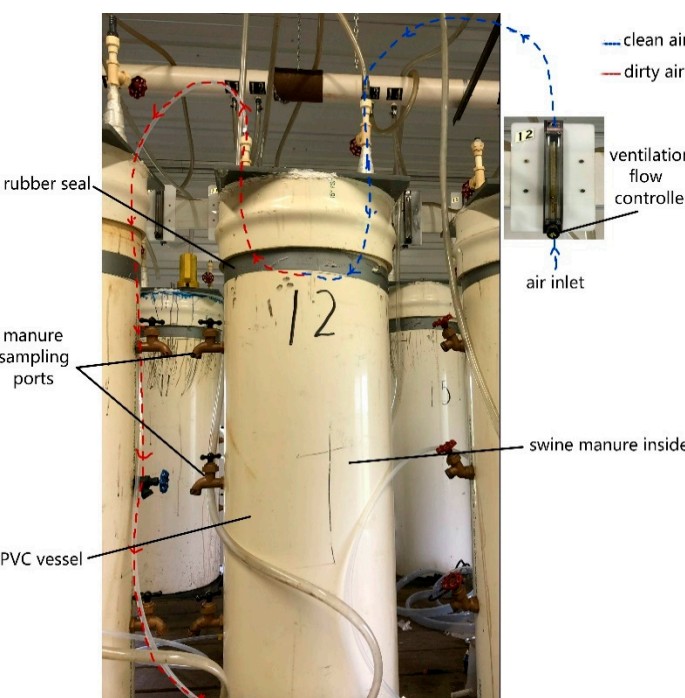

**Figure A1.** Manure simulator that simulated the deep pit storage under a slatted swine barn floor.

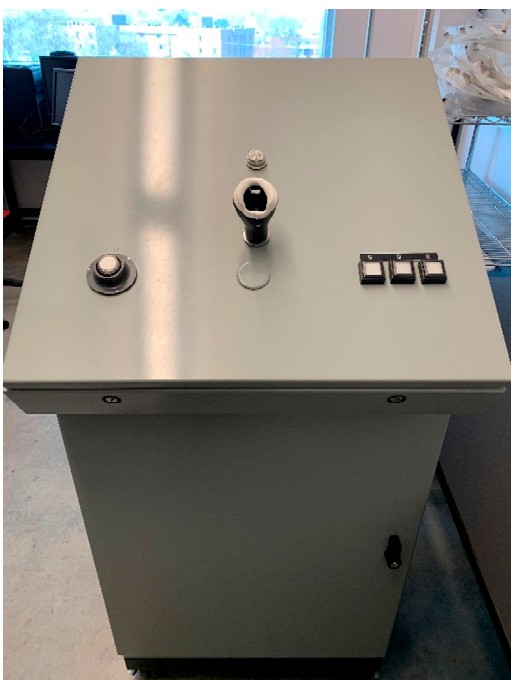

**Figure A2.** Olfactometer used to measure odor concentration.

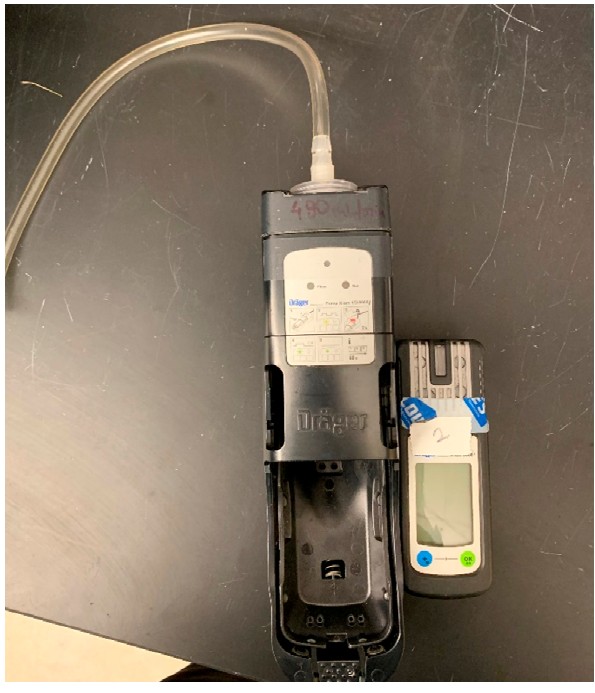

**Figure A3.** Dräeger X-5600 used to measure the concentrations of $H_2S$ and $NH_3$.

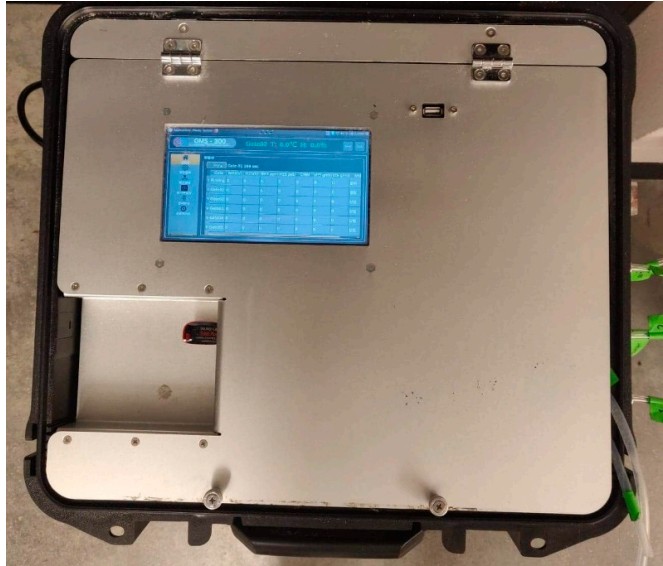

**Figure A4.** OMS-300 used to measure the concentrations of $H_2S$ and $NH_3$.

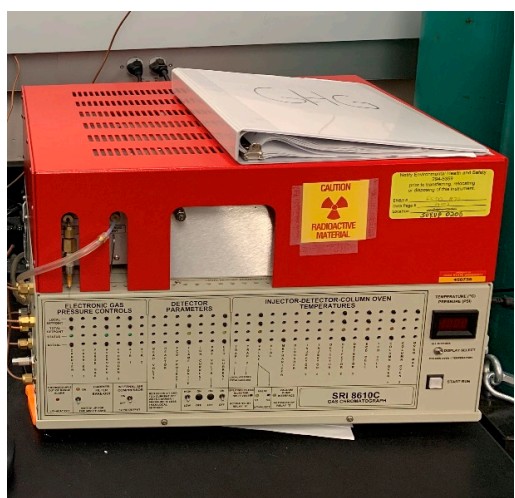

**Figure A5.** Gas chromatograph (GC) equipped with a flame ionization detector (FID) and electron capture detector (ECD) used to measure the concentrations of $CO_2$, $CH_4$, and $N_2O$.

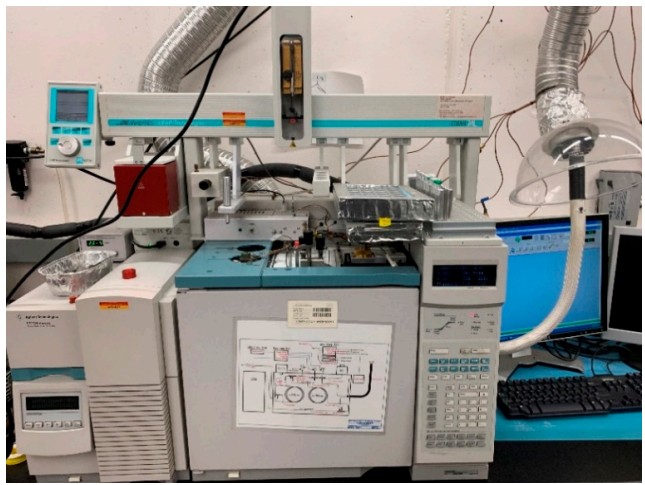

**Figure A6.** GC-MS used to measure the relative abundance of targeted VOCs.

**Table A1.** The list of manure additive products with their information.

| Name of Product | Company | Claims | Types of Additive | Recommended Application Rate |
|---|---|---|---|---|
| Triune | Agrotech | Reduce odor, solids | Large, negatively charged molecules | 1 L per 30 tons of manure |
| Manure Master Plus | ProfitProAG | Enhance manure digestion and liquefaction, reduce barn and field odors, reduce top crusting and bottom solids | Bacteria | 40 gallons (151.4 L) for a million gallon (3.78 million L) of manure, each month adding additional 2.5 gallons (9.46 L) of the product |
| Confine N | AgXplore | Decrease odor and ammonia emissions, reduce biosolids, reduce foaming. | Chemical | First treatment application is five gallons (18.9 L) of Confine N concentrate per 250,000 gallons (946,353 L) of manure. Then, add 2.5 gallons (9.46 L) of Confine N concentrate per 250,000 gallons (946,353 L) of manure monthly |
| Sulfi-doxx Dry | Direct Biologicals | Removes organic and inorganic compounds | High surface area activated carbon | 25 lbs (11.3 kg) per 1000 finishers every six months |
| Waste Away | CXI | Control odor by converting anaerobic condition into aerobic condition* | Microorganisms | 8 ounces (227 g) of the product in 20 ounces (567 g) of lukewarm water (29.4~35 °C) of the initial treatment for each of the three manure storage simulators used for this product. Then, biweekly 4 ounces (113.4 g) in 20 ounces (567 g) of lukewarm water applied as a follow-up treatment |
| More Than Manure | VERDESIAN | Reduces farm odors | Polymer | 18 ounces (510 g) per acre (4047 m$^2$) |
| Enviro Lagoon | Soutions4Earth | Heavier than water and acting where solids accumulated | Chemical | Once a week, nine gallons (34.1 L) per 1000 head swine for weeks 1 and 2, six gallons (22.7 L) per 1000 head for weeks 3 and 4, three gallons (11.3 L) per 1000 head for week 5+ |
| Oxydol | Agranco | Remediate landfills, wastewater treatment plants, and sewage | Bacteria, enzymes, and probiotics | 2 kg per 1000 pigs of the initial treatment, and then 1 kg per 1000 pigs per month |
| Sludge Away | Tomco Chemical | Removal of biological solids | Chemical | 10,000~20,000 gallons (37,854~75,708 L) of manure per three gallons (11.3 L) of the product as the initial treatment, followed by two quarts (1.89 L) for the next four weeks, and then three quarts (2.84 L) monthly |
| Penergetic g | Penergetic Solutions | Eliminates unpleasant ammonia and sulfur odors | Chemical | 3~4 lbs (1.36~1.81 kg) per (25,000 gallons) 94,635 L of manure as the initial treatment, then biweekly adding 1 lbs (0.45 kg) per 100,000 gallons (37,854 L) of manure |
| Manure Magic | Drylet | Turbo-charge the natural process of anaerobic digestion | Microbial cultures | 25 lbs (11.3 kg) per 1,000,000 gallons (3,785,411 L) of manure |
| LLMO-SST | General Environmental Science | Degrading waste and odorous compounds | Bacteria | 600 mL per 400 gallons (1514 L) of manure as the initial treatment and weekly adding 150 mL, per 400 gallons (1514 L) of manure |

* Information from the manufacturer's website (accessed on 09 June, 2020), i.e., 'Control odor by converting anaerobic condition into anaerobic condition' appears to be a typographical error.

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
