# Peer review of "Emissions from Swine Manure Treated with Current Products for Mitigation of Odors and Reduction of NH3, H2S, VOC, and GHG Emissions"

_data, 2020_

Round 1

Reviewer 1 Report

This dataset covers important data about emissions of important polluting gasses and odorants from swine manure stored with or without additives. The experimental setup seems appropriate with each time a control (without additive) next to max 3 different additives. Due to varying over 3 manure sources, however, no replicate is available for each unique treatment (additive x manure source combination). 

Although the data are considered important, however, I found some shortcomings in the paper that have to be solved before publishing this paper. Therefore, I recommend major revisions.

I found out that this paper originates from the first author's MSc thesis, which could be found at: https://lib.dr.iastate.edu/cgi/viewcontent.cgi?article=8664&context=etd

Within this thesis, the current paper is Chapter 4, with the difference that chapter 4 covers only 8 additives and this data descriptor paper 12 additives. Chapter 3, furthermore, is a research paper based on this dataset and will be submitted to Atmosphere. In the current paper I could not find any reference to this chapter or paper.

L36: No DOI is yet given (but probably this will be assigned by the journal at publication?)

L38: No license option selected

L57-58: Only product names are mentioned. Information about companies, claims, type of additive are missing. Furthermore, only "Application rate and methods of manure additives were followed by each product description." is mentioned in L145-146. This information could be added in an additional table/supplement.

L64-65: "This dataset can provide the statistic number of emissions from swine manure." What is meant by 'statistic number'? I guess you mean that from the statistical analysis of this dataset information can be retrieved on emissions from swine manure.

L93: pp, must be ppm?

L99: 'day is 0, measurements were done before any treatments'. In the dataset I found 2 measurements per trial on day = 0. What is the difference between these two?

L125-129: General columns (A-D) are explained here, whereas this information also applies for other sheets. Column E is only different for VOC (in weeks) compared to the other sheet (in days).

L141: feet, inches: non-SI units used

L143-145: Flow rate and headspace: no details on how well flow rate was controlled and how changes in air and manure temperature were accounted for. Ambient and manure temperature could influence emissions and should be mentioned for each trial.

L160-161 (Drager/OMS), L169 (GC): Details on method used and/or company names are lacking. For the GC-MS (used for VOCs), however, the description is rather elaborate.

L210: (APHA, 1998), non-numbered reference and not in reference list. And is this reference also valid for 2540 B and E (L206)?

L210: TKN: Do not use an abbreviation at the beginning of a sentence

References used in the methods section can be more specific and less self-citations.

Figure 1: How can a representative sample be guaranteed when samples are taken from the side of a tank and no mixing is applied?

There are some abbreviations not explained:

L61, 168, 260: FID / ECD

130, 199: SIM

L144: ACH

Author Response

A point-by-point response is attached.

Reviewer 2 Report

Missing information on how manure is routinely processed in the state of Iowa.

It is not explained why the Fig A1 device was used. How does the data obtained from this device differ from the data obtained from the virtually used manure processing devices??

At the same time, the text does not contain an assessment of the influence of additives on the experiments.

Data on the chemical composition of the input material and data obtained from experiments (VOC, NH3, etc.) are not evaluated. So why are they listed?

The enclosed Excel files (supplementary data) contain only basic information, with no analyte concentrations. They are virtually useless to the reader. "VOC concentrations" are not used anywhere in the text. A table with retention times, peak area and other data is useless. The reader will never know any concentration from it, although the text states that concentrations were detected by calibration (line 202-203). Only compound identification results from the enclosed Excel datasets.

Missing information on the "novelty" and benefit of the text in Introduction, which the authors call "Summary."

The text is thematically reminiscent of work in journals with IF (e.g. Cai et al., 2006), which are, however, of a very high quality level compared to the submitted text. https://www.sciencedirect.com/science/article/pii/S0021967305019977

The most problematic is the subchapter (3.4) dealing with VOC.

The information given in line 185-189 is incomprehensible to the reader, especially when using a second column. If this information is provided it is necessary to explain why restrictor and pre-column is used in the case of column 5% phenyl polysilphenylene siloxane (30 m × 0.53 mm inner diameter × 0.5µm thickness, Trajan 183 Scientific, Austin, TX, USA) and why this is not the case for column: a polar bonded polyethylene glycol column in a 184 Sol-Gel matrix x (30 m × 0.53 mm thicketer × 0.5µm thickness, Trajan Scientific column, USA). It is necessary to write and justify the use of two different phases of columns.

The row 185: The first column was fixed with a restrictor pre-column, and the mid-point of the two columns was  maintained at a constant pressure of 0.39 atm by a pneumatic switch.  Ultra-high pure He (99.999%, 187 Airgas, Des Moines, IA, USA) was used as the carrier gas. The discharge from the second column 188 was simultaneously directed to a single quadrupole Agilent mass spectrometer (MS) (Model 5973N, 189 Santa Clara, CA, USA).

Practically setting constant column pressure. What is meant by the central section of the column - is it a custom column without a restrictor and a pre-column??? What good is this information??? Why is pre-column and restrictor used? What does the other column have to do with it? It is clear from the description that the second column was connected directly to MS. Not specifying its connection to restrictor, precolumn...

Where such information is provided, it must be explained.

  • At the same time, is there no information about connecting the column to the inlet - directly or via pre-column?
  • Why is pre-column and restrictor used in the case of the non-polar column and not used in the case of the polar column?
  • How does the above connection relate to the concentration of polar and non-polar compounds in the analysed analytes from the experiments with this column arrangement in the kiln???
  • At the same time, there is no justification for using a polar and non-polar column. Attached data sets, do not contain the division of the analysed compounds into polar and non-polar.
  • Some compounds e.g. phenol - can be analysed for both polar and non-polar columns. Which column was used and from which column after measurements were phenol concentrations determined?
  • Why were two evaluation programmes used, each by a different manufacturer? Lack of justification - Figure. A reader unfamiliar with the layout of this technique cannot understand the text using the given programs. ChemStation program E.01.01.335 is currently an older, non-manufacturer-supported program.

Author Response

A point-by-point response is attached.

Round 2

Reviewer 1 Report

Most comments made to the previous version of the manuscript are dealt with properly, however, I still have some major concerns, mainly on some points not dealt with sufficiently. These points mainly deal with self-citations and temperature control / registration.

Furthermore, it would have helped me when line numbers (of the new document) of the corrections/additions would have been added in the responses.

I would like the authors to elaborate a bit on their experimental setup and especially on the lack of replicates (each additive x manure source combination is only monitored once). How different are the manure sources and would that cause different / similar effects per additive?

L70: Abbreviations should be in brackets, not the full description

My previous comment

“L125-129: General columns (A-D) are explained here, whereas this information also applies for other sheets. Column E is only different for VOC (in weeks) compared to the other sheet (in days).

Response: Added the time unit for VOCs is in weeks instead of in days.”

now L175-179 is only dealt with partially. My main point here was that a similar description is missing in earlier paragraphs where other sheets are discussed. Or at least describe it in a similar way in all places.

Concerning the response to my question about temperature:

“The measured temperature of the air in the headspace of manure was similar to the room temperature (14 to 21 deg C).  The testing room temperature was maintained by using either air conditioning or heating.”

A 6 deg C difference could considerably influence emissions from manure. So my question is still how this variation could have influenced the results. Is it day-night fluctuation or are some periods / experiments done under warmer/cooler conditions than others? This information is rather essential when interpreting the data.

L253-257: Information about the GHG GC is lacking (I mentioned this previously for NH3/H2S method, which is now added).

Concerning my point about self-citations. It is still not clear to me why so many references are needed in the methods section (grouped at the end of the paragraph). You should not mention all your (groups) work which used similar methods, but just 1 or 2 references that describe the method (and probably some modifications in the second reference) appropriately. In this way it will become very difficult for the reader to find out the details about the method. If no proper single reference is available, description in this paper should be sufficiently clear.

L400; Table A1: Rather difficult to read due to track changes, so needs to be checked thoroughly. Some questions/remarks:

  • Please convert all units to SI units
  • Most claims are rather vague, like ‘organic and inorganic compounds’, ‘acting’, ‘remdiate’, ‘bio(logical) solids’, ‘turbo-charge’, ‘waste and odorous compounds’, but this is probably how the companies state it. Detailed information about (claimed) working mechanism is probably often missing. When available, please add this.
  • MMP, application rate: “is 40 gallons …. ” Why “is”?
  • WA, claim: “converting anaerobic condition into anaerobic condition”, twice “anaerobic”
  • More than manure, appl.rate: “per acre”?, how converted to volume unit?

Reviewer 2 Report

Comments were answered and the information requested was added to the text. I agree with the publication in the journal.

Author Response

Thank you for the feedback. 

Round 3

Reviewer 1 Report

Thanks for revising the manuscript. The manuscript has considerably improved.

For me, the 'track changes' presentation of revisions makes it not very clear. Especially when changes are made by different authors and both rounds of revisions are cumulatively stored in the manuscript it becomes rather 'messy'. I would prefer starting from the previously submitted version, which is the version the reviewer commented on, and adding revisions and marking them with the yellow marker. Furthermore, the line numbers referred to should be checked by the authors after conversion to pdf, as presentation in word will often differ from site by site.

Maybe this way of marking revisions (track changes) is prescribed by the journal, so I will make a similar comment to the editor.

With respect to the manuscript itself, only one minor point remains. You end the response about the replicates with: "We added the new commentary about replicates and additives per manure in Lines 180-186."

My response to this is: In the pdf version I could download it seems to be L252-258. In the first sentence “range vary” seems to be double. Please correct. L254: “working manure” should be “working with manure”?
